# Nutritional Approaches in Neurodegenerative Disorders: A Mini Scoping Review with Emphasis on SPG11-Related Conditions

**DOI:** 10.3390/nu17213344

**Published:** 2025-10-24

**Authors:** Paulo Renato Ribeiro, Carmen Ferreira, Carlos Antunes, Gonçalo Dias, Maria João Lima, Raquel Guiné, Edite Teixeira-Lemos

**Affiliations:** 1ESAV, Polytechnic University of Viseu, 3500-606 Viseu, Portugal; p_renato_saraiva_r19@hotmail.com; 2CERNAS-IPV Research Centre, Polytechnic University of Viseu, 3500-606 Viseu, Portugal; mjoaolima@esav.ipv.pt (M.J.L.); raquelguine@esav.ipv.pt (R.G.); 3ESAS, Polytechnic University of Santarém, 2001-904 Santarém, Portugal; carmensousaferreira@gmail.com; 4CERNAS-IPC Research Centre, Polytechnic of Coimbra, 3045-601 Coimbra, Portugal; carlosalbertoantunescb@gmail.com (C.A.); gdias44@hotmail.com (G.D.); 5ESACB, Polytechnic University of Castelo Branco, 6001-909 Castelo Branco, Portugal; 6APROSER, Association of Producers from Sertã, 6100-601 Sertã, Portugal; 7ESAC, Polytechnic University of Coimbra, 3045-601 Coimbra, Portugal

**Keywords:** genetic factors, dietary interventions, metabolic responses, SPG11, spatacsin, quality of life, neuronutrition, microbiome modulation, precision medicine, multi-omics

## Abstract

**Background:** Neurodegenerative diseases, including spastic paraplegia type 11 (SPG11), are complex disorders characterized by progressive neurological decline and significant metabolic disturbances. Spatacsin, the protein encoded by the SPG11 gene, plays a critical role in autophagy and lysosomal homeostasis, which are essential for neuronal health. Its impairment leads to defective cellular clearance and neurodegeneration. Recently, personalized and precision nutrition have emerged as promising approaches to enhance clinical outcomes by tailoring dietary interventions to individual genetic, metabolic, and phenotypic profiles. **Objectives:** This mini scoping review aimed to synthesize current evidence on the application of personalized and precision nutrition in SPG11 and to explore how insights from related neurodegenerative diseases could inform the development of future dietary and metabolic interventions for this rare disorder. **Methods:** Following PRISMA-ScR guidelines, a scoping review was conducted using PubMed, Scopus, and Web of Science databases (2020–2024). Eligible studies included investigations addressing nutritional, genomic, or metabolic interventions in neurodegenerative diseases. Of 30 screened papers, nine met the inclusion criteria, primarily focusing on nutritional and metabolic interventions related to neurodegenerative and neuromuscular conditions. **Results:** To date, no dietary intervention trials have been conducted specifically for SPG11. However, evidence from studies on related neurodegenerative diseases suggests that antioxidant, mitochondrial-supportive, and microbiota-targeted dietary approaches may beneficially influence key pathological processes such as oxidative stress, lipid dysregulation, and autophagy—core mechanisms that are also central to SPG11 pathophysiology. **Conclusions:** Although current evidence remains preliminary, personalized nutrition is a promising supplementary strategy for managing neurodegenerative diseases, including SPG11. Future research should incorporate systems-based approaches that combine dietary, metabolic, and neuroimaging assessments, with sex and comorbidity-stratified analyses, multi-omics profiling, and predictive modeling. These frameworks could help design safe, effective, and personalized nutritional interventions aimed at enhancing metabolic resilience and slowing disease progression in SPG11.

## 1. Introduction

Genetic factors are a crucial determinant of an individual’s health profile and susceptibility to neurodegenerative diseases [1]. Precision nutrition aims to personalize dietary plans based on genetic, phenotypic, and metabolic traits, along with nutritional requirements and epigenetic influences [2,3]. This area of nutritional science recognizes the diversity of neurodegenerative disorders, including rare conditions, by customizing dietary strategies to each person’s biological profile. These approaches have the potential to improve disease management by targeting disease mechanisms and reducing symptoms [1,4].

Hereditary spastic paraplegia (HSP) is a diverse group of neurodegenerative disorders that mainly impact upper motor neurons, causing progressive spasticity and gait problems. Among them, spastic paraplegia type 11 (SPG11) results from a mutation in the SPG11 gene [5]. This genetic mutation affects the production of spatacsin, a large membrane protein mainly found in neuronal tissues and crucial for cellular processes essential to neuronal health [6,7]. It is involved in autophagic and lysosomal pathways that mediate the degradation and recycling of damaged organelles and misfolded proteins, thus maintaining cellular homeostasis [6,7,8]. It also contributes to processes such as axonal growth, synaptic maintenance, and mitochondrial integrity, all of which are essential for neuronal maturation and function. Its involvement in these pathways highlights the importance of spatacsin in supporting normal neurodevelopment and neuronal survival [5].

Conversely, mutations that cause the loss of spatacsin function disrupt autophagic and lysosomal pathways, leading to the accumulation of defective organelles and toxic protein aggregates within neurons. This dysfunction triggers cellular stress, mitochondrial impairment, and ultimately neurodegenerative pathways. Evidence from animal models, including zebrafish and mice, supports spatacsin’s critical role in neuronal health; its deficiency leads to impaired motor neuron development and neurodegeneration [9]. Furthermore, the widespread expression of spatacsin across multiple tissues helps explain the multisystem nature of SPG11, which typically presents with motor impairments, spasticity, urinary dysfunction, and systemic features [10]. Thus, spatacsin functions as a bifunctional protein, essential to both neurodevelopment and neurodegeneration. Its loss results in the progressive neurological decline characteristic of SPG11 [8,11].

From a clinical perspective, SPG11-related neurodegeneration significantly affects motor function, coordination, and mobility, particularly in the lower limbs [10]. Individuals affected by the condition often experience weakness and increasing lower limb spasticity, along with characteristic gait disturbances that involve a reduced range of motion caused by spasms and spastic co-contraction during voluntary movement, as well as static postural issues abnormalities [12,13]. Additional symptoms may include urinary dysfunction, impaired balance, and decreased vibratory sensation [5], emphasizing the disorder’s multisystem nature.

Despite its rarity and variability, SPG11 shares common pathogenic mechanisms with other neurodegenerative diseases such as Alzheimer’s and Parkinson’s, including mitochondrial dysfunction, oxidative stress, inflammation, and alterations in lipid metabolism and gut microbiota [14,15]. This overlap underscores the need for mechanism-based and targeted interventions, including emerging nutritional strategies that may modulate these biological pathways [16,17].

Recently, the concept of neuronutrition has gained significant attention as a new field within nutritional systems biology [18]. This area emphasizes the importance of personalized approaches and tailored dietary plans to meet individual nutritional needs [19]. Advances in “omics” technologies and bioinformatics have revolutionized nutrition by enabling personalized interventions that transcend traditional one-size-fits-all recommendations. This shift toward individualized treatments enables the identification of biomarkers—such as those derived from genomics, metabolomics, and microbiomics—that reflect disease-specific pathogenic processes [20]. Utilizing these insights, dietary interventions can be tailored to improve outcomes. For instance, ketogenic diets can enhance mitochondrial energy production, antioxidant-rich diets can reduce oxidative stress, and the microbiota can be modulated through the use of prebiotics and probiotics. Such precision strategies aim to target disease pathways at multiple levels, potentially slowing neuronal degeneration and alleviating symptoms more effectively than generic approaches. Although direct evidence for SPG11 is still limited, research on related neurodegenerative disorders suggests that personalized nutritional strategies may target core pathogenic mechanisms, helping reduce symptoms and improve quality of life.

Given the limited and still-emerging literature specifically addressing personalized nutritional strategies for SPG11, a full systematic review may not yet be feasible or necessary. Instead, this study adopts a mini-review format to provide a focused and timely summary of the current evidence related to precision nutrition in neurodegenerative and neuromuscular disorders with similar pathogenic mechanisms, emphasizing emerging strategies in neuronutrition. This focused overview summarizes key findings, highlights significant insights, and identifies knowledge gaps that can inform future research and clinical applications.

Finally, this mini scoping review explores the role of precision nutrition as a targeted approach to improve health and clinical outcomes for individuals with SPG11 and related disorders. By exploring current knowledge of nutritional needs, the impact of diet on neurodegenerative disease, individual responses to dietary interventions, and available clinical evidence, this review underscores the potential of personalized nutritional strategies to enhance both quality of life and functional outcomes.

## 2. Methodology

This work was developed as a scoping mini-review, an approach particularly suited to emerging research areas where the available evidence remains limited or fragmented [21]. Such reviews allow researchers to synthesize what is currently known, highlight key themes and trends, and identify gaps that may guide future investigations. Although not a full systematic review, this study followed the structure and transparency principles of the PRISMA and PRISMA-ScR guidelines [22]. These frameworks were adapted to the exploratory nature of our study to ensure clarity, reproducibility, and rigor while preserving the flexibility that a scoping approach requires.

The main objective of this review was to gather and contextualize current evidence on precision nutrition strategies relevant to spastic paraplegia type 11 (SPG11) and related neurodegenerative conditions. Given the distinctive genetic and metabolic characteristics of SPG11, particular emphasis was placed on understanding how personalized nutritional interventions may influence disease mechanisms and support clinical management.

### 2.1. Literature Search

A comprehensive search was conducted on 24 March 2025, using three major scientific databases—Web of Science, PubMed, and Scopus—to ensure broad coverage and diversity of sources. Each database was chosen for its unique strengths. PubMed offers extensive biomedical literature, especially on clinical trials and translational research related to neurodegenerative diseases and nutritional interventions. Web of Science provides high-quality indexing of peer-reviewed journals and conference proceedings, along with citation-tracking features that help assess impact and relevance. Scopus was included for its wide content coverage and advanced analytical tools, which support research connecting diet to genetics and metabolic health.

An advanced search string was constructed to capture the main concepts of precision nutrition, disease classification, and omics-based research: (“precision nutrition” OR “personalized nutrition” OR “tailored nutrition” OR “targeted nutrition”) AND (“spastic paraplegia type 11” OR “SPG11” OR “hereditary spastic paraplegia” OR “HSP” OR “neurodegenerative disease” OR “neuromuscular disorder” OR “neurodegeneration”) AND(“nutrigenomics” OR “genomic nutrition” OR “metabolic phenotypes” OR “genetic factors” OR “microbiome” OR “microbiota” OR “metabolomics”).

This search string was carefully designed to include a broad range of related terms, capturing studies that may use different terminology or focus on slightly different aspects of the topic. This systematic approach to database selection and search formulation ensured the retrieval of studies addressing personalized nutritional strategies incorporating genetic and metabolic factors in neurodegenerative diseases. The goal was to obtain a representative and comprehensive overview of the current research landscape to support informed conclusions and recommendations.

### 2.2. Inclusion Criteria

Several restrictions were applied to the literature search to ensure the results were current, reliable, and relevant. Only articles published from 2020 onward were included, so the review would reflect the latest developments in the field [23]. Eligible studies had to be written in English and relate to research areas relevant to this topic, including neuroscience, neurology, nutrition and dietetics, hereditary genetics, food science and technology, experimental medicine, and nutrient-related research [24].

In the subsequent stage, the search was expanded to include peer-reviewed original research articles and systematic reviews from a wide range of reputable scientific journals known for publishing studies in this field [25]. The inclusion criteria were further broadened to incorporate full-text articles accessible through institutional subscriptions or other legitimate access methods, ensuring comprehensive coverage and enabling a critical evaluation of existing literature. This approach facilitated a deeper understanding of the current state of research, while also allowing the identification of emerging themes and future research opportunities [23,24]. To ensure methodological transparency and reliability, only open-access or publicly available articles were retained for full-text analysis. Duplicates and ineligible studies identified during preliminary screening were removed. The final selection included both empirical and review articles, providing a comprehensive synthesis of current evidence and minimizing potential sources of bias.

### 2.3. Article Selection Process

The article selection process followed a structured and transparent procedure guided by the PRISMA and PRISMA-ScR frameworks, which were adapted for the exploratory nature of this scoping review. Although not intended to be a full systematic review, these frameworks ensured methodological rigor and reproducibility. The aim was to create a focused synthesis of the most recent and relevant literature on personalized nutrition in neurodegenerative and neuromuscular diseases, especially concerning SPG11.

Comprehensive searches were conducted in PubMed, Scopus, and Web of Science. Differences in the number of records retrieved from each database—for example, the smaller yield from Scopus compared with Web of Science—were attributed to variations in indexing depth and keyword mapping for specific terms such as “SPG11,” “precision nutrition,” and “nutrigenomics.” To maintain transparency and data accessibility, only peer-reviewed, open-access studies were included in the final selection. While this introduces some limitations compared with a full PRISMA-compliant process, it allowed the research team to perform a complete and consistent evaluation of all eligible articles.

After applying the predefined restrictions, all records were exported to a centralized Excel spreadsheet for organization and screening. Data were extracted from each article on study type, authors, title, journal, keywords, abstract, publication year, and DOI. Screening was conducted based on titles, abstracts, and keywords to determine eligibility. Studies were included if they addressed neurodegenerative or neuromuscular diseases, hereditary spastic paraplegia (including SPG11), precision nutrition, nutrigenomics, or genomic and metabolic factors relevant to dietary interventions. Articles were excluded if they did not cover these topics or if the study design did not align with the review’s objectives.

The initial screening was carried out independently by three members of the research team. When uncertainty arose, a fourth reviewer joined the discussion to reach a consensus. The remaining studies were then reviewed in full, and additional exclusions were made when content did not meet the inclusion criteria.

A total of 4067 articles were identified during the initial search. No duplicates were found. A publication date filter was applied, limiting the results to articles published from 2020 onward, which reduced the count to 1861 articles (2206 were excluded). Filters based on relevant research areas were then applied, resulting in 1533 articles (328 were excluded). After selecting journals that published studies related to the topic, 1159 articles remained (374 were excluded). Only articles published in English were kept, resulting in 1144 articles (15 were excluded). Finally, only review articles were included, bringing the total to 223 articles (921 were excluded).

The four members of the research team independently screened the titles, abstracts, and keywords of the 223 articles and excluded 132 of them based on the eligibility criteria established in the study’s methodology. After reviewing the full text of the remaining 91 articles, 61 were excluded due to lack of open access, which prevented comprehensive analysis. As a result, 30 articles were read and analysed in detail. Of these, one was excluded because its research objective did not align with the focus of this review, and fourteen were excluded because they did not address nutritional interventions in neurodegenerative diseases. Consequently, a total of 9 articles were included in this mini review.

A flow diagram summarizing the identification, screening, and inclusion process is presented in Figure 1.

## 3. Results

Of the 30 articles initially identified, only nine met the inclusion criteria and were incorporated into this scoping mini review. Analysis of these studies revealed several approaches that integrate personalized nutritional strategies with the underlying genetic and pathophysiological mechanisms of degenerative neuromuscular diseases. Importantly, no nutritional interventions were found that were specifically designed for individuals with SPG11.

However, multiple studies investigating precision nutrition in other neurodegenerative and neuromuscular disorders were identified. These works provide a valuable foundation that can be adapted to support individuals with SPG11, while taking into account their distinct genetic, metabolic, and microbiota profiles [5,7].

Table 1 summarizes the key findings from these studies and highlights their relevance to potential dietary strategies for SPG11. The table provides an overview of emerging evidence supporting the therapeutic potential of nutritional interventions, particularly those that emphasize antioxidant, anti-inflammatory, and mitochondrial-supportive mechanisms in neurodegenerative disease. Collectively, findings from related conditions—such as Alzheimer’s disease, Parkinson’s disease, multiple sclerosis, amyotrophic lateral sclerosis, and traumatic brain injury—suggest that targeted dietary modulation may offer complementary benefits for managing metabolic and neurodegenerative dysfunction in SPG11 [3,14,26].

The evidence gathered in this scoping mini-review predominantly relates to neurodegenerative diseases such as Alzheimer’s, Parkinson’s, and multiple sclerosis. However, there remains a marked absence of direct clinical or experimental studies specifically addressing nutritional interventions in individuals with SPG11.

Among the nine studies analyzed, none were exclusively designed for SPG11 patients. Nevertheless, the work of Berciano et al. [2] and Abeltino et al. [4] highlights emerging initiatives and the potential of tailored nutritional strategies that account for genetic, metabolic, and microbiota profiles in SPG11 and related hereditary spastic paraplegias. These studies suggest that personalized, data-driven approaches integrating genomic and metabolic information could improve disease management, although definitive clinical validation is still lacking.

Considering this gap, the present review draws upon findings from other neurodegenerative and neuromuscular disorders to guide future strategies for SPG11. Interventions such as antioxidant supplementation, mitochondrial support, and adherence to dietary patterns like the Mediterranean diet have demonstrated neuroprotective effects in related conditions [14,26]. Nonetheless, such findings should be considered preliminary when applied to SPG11, underscoring the need for dedicated research efforts. Considering the biological variability characteristic of SPG11 and other neurodegenerative diseases, future personalized dietary plans must incorporate each patient’s unique genetic, metabolic, and microbiota profile.

## 4. Discussion

This study highlights the promising role of personalized nutrition as a crucial strategy for managing neurodegenerative diseases, including rare conditions such as SPG11. Although direct clinical evidence specifically addressing SPG11 remains limited, the extensive range of reviewed studies offers compelling insights into the potential of individualized nutritional interventions to influence disease pathways and enhance patient outcomes.

### 4.1. Direct Clinical Evidence

A key theme in the literature is the inherent variability of neurodegenerative diseases. Disorders such as Alzheimer’s, Parkinson’s, multiple sclerosis, and hereditary spastic paraplegias have complex genetic, metabolic, and microbial profiles. This variability requires a shift from broad dietary guidelines to personalized nutritional strategies that take into account each patient’s unique biological characteristics [14,24]. Clinical and meta-analytic evidence supports this approach. For example, interventions like ketogenic diets, caloric restriction, and supplementation with antioxidants or essential fatty acids have shown different levels of success across various populations [1,29]. Meta-analyses further confirm that biomarker-guided nutritional strategies can mitigate disease progression and enhance quality of life [27]. These findings provide direct clinical evidence that individualized nutrition can positively influence neurodegenerative outcomes Importantly, these studies also reveal the impact of individual genetic vulnerabilities, metabolic states, and gut microbiota composition, all of which influence disease progression and response to dietary changes [27]. Such heterogeneity underscores the rationale for extending personalized nutrition research to rarer neurodegenerative disorders, including SPG11.

### 4.2. Mechanistic Extrapolation

While direct SPG11-specific data are lacking, mechanistic insights from related diseases provide a strong biological basis for personalized nutritional interventions.

Personalized nutritional approaches based on genetic and microbiota profiles enable targeted modulation of neuroinflammatory and neurodegenerative pathways, potentially slowing disease progression and reducing symptoms The work of Singar et al. [1] illustrates how customizing nutritional interventions based on individual genetic and microbiota profiles can lead to more effective management of neurodegeneration by directly targeting mitochondrial function, oxidative stress, and neuroinflammation—core features of SPG11 pathology [1].

Diets rich in bioactive compounds such as those characteristics of the Mediterranean and Atlantic patterns, have been shown to promote neuroprotection through modulation of lipid metabolism and inflammatory responses which are major hallmarks of neurodegenerative disease [26,28,30]. These mechanisms suggest that dietary changes could help slow SPG11 progression. Moreover, such diets may allow patient stratification into subgroups with distinct nutritional responses, reducing variability and improving therapeutic results [1,3].

Beyond the well-known benefits of the Mediterranean and Atlantic dietary patterns, naturally rich in omega-3 fatty acids, polyphenols, and antioxidants, other plant-derived bioactive compounds may also enhance their neuroprotective potential. For instance, glucosinolate-derived isothiocyanates from Brassicaceae vegetables [31], pectin-rich citrus extracts [32], and conifer by-products obtained through hydrodynamic cavitation [33] have demonstrated promising neuroprotective effects. These compounds are believed to activate Nrf2 pathways and reduce oxidative stress, inflammation, and lipid peroxidation—mechanisms also linked to SPG11 pathophysiology. Because these bioactives are part of Mediterranean and Atlantic diets, they could complement nutritional strategies aimed at reducing oxidative and lysosomal stress.

Although direct studies in SPG11 are still lacking, these findings open new avenues for future nutritional research in this condition.

Additionally, research from both animal models and human studies indicates that dietary changes affecting the microbiota and metabolic pathways can significantly impact neurodegenerative processes. The gut–brain axis has become a key focus for personalized nutritional interventions. Recent studies emphasize the importance of this brain–gut axis—an interactive, bidirectional communication network involving neuroimmune, neuroendocrine, and neural pathways—in influencing neurodegeneration and motor function in SPG11 [34]. By analyzing the composition of individual microbiota, we can create dietary plans that foster beneficial microbial communities and metabolic environments supportive of neuroprotection. Cohen et al. [29] highlighted the potential of using diet to modulate microbiota as an indirect therapeutic approach to neurodegeneration. Together, these mechanisms illustrate how diet influences neurological health beyond basic nutrition—through systemic regulation of inflammation, metabolism, and cellular resilience. Although these advances are encouraging, significant challenges remain in applying personalized nutrition in clinical practice.

### 4.3. Relevance of Sex, Diet, and Comorbidities in SPG11

The complexity of neurodegenerative diseases and their variable responses to treatment emphasize the need for thorough clinical, genetic, and metabolic assessments [35,36]. Sex differences, dietary habits, and metabolic comorbidities strongly influence disease progression and therapeutic outcomes.

Evidence shows that SPG11 affects not only the nervous system but also metabolism. In a cross-sectional study, Regensburger et al. [37] found that patients with SPG11 had a higher body fat index, lower lean mass and muscle volume, altered adipokine levels (leptin, resistin, progranulin), and reduced hypothalamic volume compared with controls. These findings suggest a link between obesity and hypothalamic neurodegeneration and indicate a systemic metabolic phenotype accompanying SPG11.

However, sex-specific metabolic responses in SPG11 remain unknown. Although the Regensburger cohort included equal numbers of men and women, no sex-based analysis of metabolic markers was performed. Broader literature shows that sex affects metabolic homeostasis, obesity risk, adipokine signaling, and lipid metabolism [38]. In SPG11, comorbidities such as obesity, reduced mobility, and adipokine dysregulation appear integral to disease expression, with progressive motor impairment driving immobility, increased adiposity, inflammatory imbalance, and worsening metabolic stress [37]. Elevated leptin and resistin levels may indicate leptin resistance and impaired energy regulation, while reduced hypothalamic volume points to central dysfunction in energy homeostasis and metabolic control [37].

The Composite Dietary Antioxidant Index (CDAI which integrates dietary intake of vitamins A, C, and E, magnesium, selenium, and zinc has been inversely associated with cardiovascular and metabolic risk [35]. The relationship between CDAI and lipid profile also appears to vary by sex, likely due to hormonal effects on oxidative balance and lipid metabolism. Given that oxidative stress and lysosomal dysfunction drive SPG11 neurodegeneration, sex-specific optimization of antioxidant and lipid-modulating nutrients is biologically plausible. For instance, men—who often exhibit higher baseline oxidative stress—may benefit more from antioxidant-focused strategies, whereas women may respond better to nutrients that support lipid metabolism and complement estrogen’s metabolic effects.

Because lipid metabolism impairment is central to SPG11 pathology (autophagy–lysosomal dysfunction with lipid accumulation) and obesity is common, dietary strategies emphasizing healthy lipid profiles, caloric balance, and antioxidant support could be particularly useful. Nevertheless, the lack of sex-stratified clinical data in SPG11 limits firm conclusions and underscores the need for targeted studies.

### 4.4. Translational and Practical Considerations

Although current data specifically focused on SPG11 populations remain limited, findings from other well-studied neurodegenerative models provide a valuable foundation for translational research. Regensburger et al. [37] demonstrated that dietary modulation can influence neurodegenerative processes by altering microbiota composition and metabolic activity. These results suggest that nutritional interventions could play an active role in disease modulation rather than serving only as supportive care.

Incorporating variables such as sex, lipid profiles, and comorbidities is essential, because these factors likely shape both the disease trajectory and the individual response to nutritional therapy. Understanding these interactions strengthens the rationale for developing dietary interventions tailored to each patient’s genetic background, metabolic status, and microbiota profile. Emerging technologies—including advanced microbiota profiling, metabolic phenotyping, and AI-based predictive modeling [3] are expected to accelerate the implementation of precision-nutrition strategies by allowing clinicians to identify responsive subgroups and optimize interventions in real time.

Translating these tools into clinical practice will, however, require overcoming several challenges. Ethical considerations, logistical limitations, and economic constraints must all be addressed to ensure that precision nutrition can be applied safely and equitably. Developing affordable diagnostic technologies, harmonized clinical protocols, and fair-access frameworks will be crucial steps toward making personalized nutrition a realistic component of neurodegenerative disease management. When achieved, these advances may help slow disease progression, alleviate symptoms such as spasticity, and improve patients’ quality of life [14].

To guide this translational effort, Figure 2 presents a proposed research roadmap for implementing precision nutrition in SPG11. The framework outlines sequential yet overlapping phases, from preparatory work to clinical application. The preparatory stage focuses on protocol design, ethics approval, registry establishment, and training of clinical teams. The pilot phase then evaluates the safety, feasibility, and adherence of tailored diets—such as ketogenic or calorie-restricted regimens—in a small SPG11 cohort. Next, integration of multi-omics and artificial-intelligence tools enables refinement of biomarkers and identification of metabolic targets. The active-trial phase expands these efforts through multicenter implementation and longitudinal monitoring of clinical, metabolic, and microbiota outcomes. Throughout all stages, ethical oversight, patient engagement, and data transparency are maintained to ensure safe and equitable translation into routine clinical practice.

The figure illustrates the progressive steps involved in implementing precision nutrition approaches for SPG11, moving from protocol design and pilot testing to multi-omics integration and active clinical trials. The framework highlights key priorities such as feasibility assessment, biomarker validation, and the ethical translation of research findings into clinical practice.

### 4.5. Strengths and Limitations

This mini scoping review has several strengths that enhance its contribution to understanding personalized nutrition in neurodegenerative diseases, particularly SPG11. First, it applies a comprehensive and transparent methodology, following PRISMA guidelines, which ensures clarity and reproducibility in the analysis. The systematic search of major databases (Web of Science, PubMed, Scopus), along with clear inclusion and exclusion criteria, reduces bias and broadens the evidence base. By focusing on publications from 2020 onward, the review captures recent advances in precision nutrition, genomics, and neurodegeneration.

Second, the review’s integrative approach—considering genetic, metabolic, microbiota, and dietary factors—provides a holistic understanding of personalized nutrition. This approach aligns with current models of individualized care, which are especially relevant for heterogeneous and rare conditions like SPG11. Importantly, priority was given to peer-reviewed original studies and systematic reviews published in reputable journals, enhancing the reliability of the evidence and providing a strong foundation for future research and clinical practice.

Despite these strengths, the review has some limitations. Few studies have directly investigated nutritional interventions tailored for SPG11, so much of the discussion extrapolates findings from related neurodegenerative diseases. Consequently, the applicability of these insights to SPG11 may be limited by its distinct genetic and phenotypic features. Moreover, because the review relies on previously published data rather than new empirical research, its conclusions depend on the quality and scope of existing studies. Finally, limiting the search to English-language and open-access publications, while practical, may have excluded additional relevant evidence, slightly reducing the overall comprehensiveness of the review.

## 5. Conclusions and Future Perspectives

This scoping mini-review emphasizes the increasing importance of personalized nutrition as a potential complementary strategy for managing neurodegenerative diseases, including the rare hereditary spastic paraplegia SPG11. Although there are currently no direct clinical or interventional studies in SPG11, growing evidence from related neurodegenerative and metabolic disorders supports the idea that customized nutritional approaches may positively impact disease mechanisms central to SPG11 pathology—such as mitochondrial dysfunction, lipid buildup, oxidative stress, and impaired autophagy. Current findings indicate that SPG11 should be considered a multisystem disorder, extending beyond neurodegeneration to include systemic metabolic alterations. These alterations including elevated adipokine levels, increased adiposity, and hypothalamic atrophy, as described by Regensburger et al. [37], suggest that energy homeostasis and lipid regulation are directly implicated in disease progression. This perspective is more systemic of the disease opens new opportunities for nutritional and metabolic interventions aimed at modulating these dysfunctions.

Looking ahead, advancing the field of personalized nutrition in SPG11 requires a translational, multi-omics approach.

Future research should focus on longitudinal studies in SPG11 that include detailed dietary assessments, metabolic biomarkers, and neuroimaging data. Analyses stratified by sex and comorbidities are necessary to clarify how hormonal and metabolic factors influence dietary responses and disease progression. Using multi-omics approaches that combine genomic, metabolomic, and microbiomic profiling could help identify modifiable nutritional targets related to neurodegeneration. Adding AI-driven predictive models may improve the accuracy and clinical usefulness of personalized dietary strategies. It is also important to address the ethical, practical, and economic aspects of implementing personalized nutrition in rare diseases through frameworks that promote accessibility, affordability, and scalability. Collaboration among neurologists, nutritionists, geneticists, and data scientists will be essential for developing and validating evidence-based, personalized nutrition protocols. Education and training programs for clinicians and patients are equally important to successfully translate emerging research into practical clinical applications.

In summary, although empirical evidence specific to SPG11 remains limited, the convergence of metabolic, microbiota, and nutritional research provides a promising foundation for future translational studies. Precision nutrition anchored in systems biology and supported by integrative biomarker profiling holds real potential to enhance metabolic homeostasis, promote neuroprotection, and improve the quality of life in individuals affected by SPG11.

## Figures and Tables

**Figure 1 nutrients-17-03344-f001:**
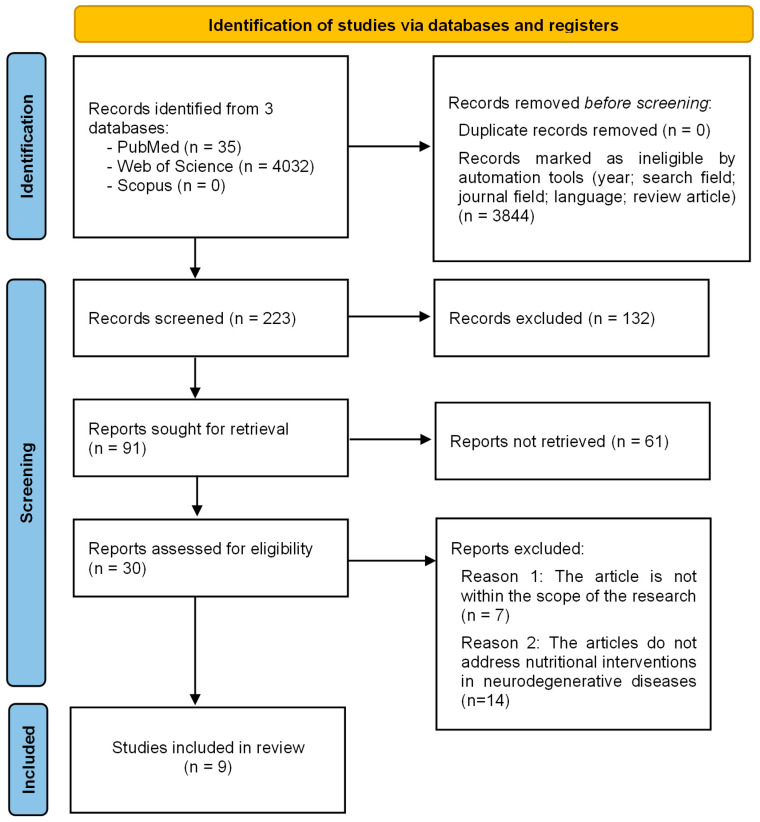
Flow chart indicating the article selection process according to the Preferred Reporting Items for Systematic Reviews and Meta-Analyses (PRISMA) flowchart.

**Figure 2 nutrients-17-03344-f002:**
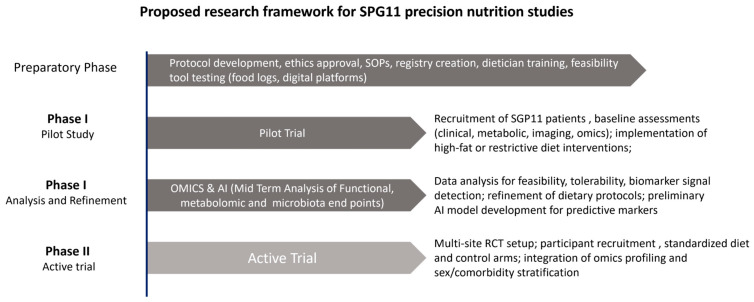
Proposed translational research framework for precision nutrition in SPG11.

**Table 1 nutrients-17-03344-t001:** Summary of studies addressing precision and personalized nutrition strategies with potential relevance to SPG11.

Study	Key Findings	Relevance to SPG11	Type of Intervention
[4]	Assessed the impact of the Italian Mediterranean diet on gut microbiota and metabolic health in individuals with metabolic syndrome. Found improvements in beneficial bacteria, SCFA production, insulin sensitivity, and reduction in inflammation.	While not directly related to SPG11, emphasizes the importance of gut microbiota and diet in neurological and metabolic health, potentially applicable in managing neurodegenerative conditions like SPG11.	Dietary intervention; 6-month Mediterranean diet adherence, microbiota analysis, metabolic parameters
[2]	Overview of precision nutrition using omics, AI, and wearable tech to provide personalized dietary guidance. Discussed challenges such as validation, ethics, and accessibility.	Potential applicability in personalized management of SPG11 patients, tailoring diet based on genetics, microbiome, and lifestyle.	Review of technologies and data types; no specific intervention; conceptual discussion
[3,7]	Focused on nutrient roles (calcium, vitamin D, magnesium, B-vitamins) and AI-powered personalized nutrition. Highlighted nutrient importance in muscle, nervous system, and disease management.	Nutritional support may benefit neurodegenerative features in SPG11, especially motor and neurological symptoms. AI-based personalized plans could optimize individual needs.	Nutritional supplementation; personalized plans via AI; data review and hypothesis
[27]	Discussed precision nutrition in public health, shifting focus from deficiency prevention to chronic disease reduction via personalized dietary recommendations, considering population heterogeneity.	Highlights the importance of tailored dietary management in complex neurodegenerative diseases like SPG11, where uniform guidelines may be insufficient.	Review; emphasizes research and technological development needed for precision nutrition
[14,26]	Presented Mediterranean diet’s neuroprotective effects, including improved metabolic profiles, gut microbiota, and cognitive function.	Med diet components could support neuroprotection and metabolic health in SPG11; potential adjunct therapy for symptom management.	Dietary intervention; long-term diet adherence and analysis
[28]	Discussed specific diets like ketogenic, anti-inflammatory, and Atlantic diet, showing effects on neurodegenerative disease markers, inflammation, and gene expression.	Potential dietary strategies to mitigate neurodegeneration and inflammation in SPG11. Fits into personalized and anti-inflammatory approaches.	Varied diet interventions; clinical and molecular data
[29]	Focused on personalized nutrition powered by AI, integrating microbiome data, dietary patterns, and responses, enabling tailored dietary management.	Represents a promising approach for personalized management of SPG11, optimizing interventions based on individual microbiome and response profiles.	AI-driven personalized nutrition; microbiome and dietary data integration
[3,26]	Low-carb, high-fat diet inducing ketosis, supporting neurodegenerative disease management. Benefits include reduced inflammation, improved mitochondrial and cognitive function, Moderate caloric intake recommended.	The ketogenic diet may offer neuroprotective benefits, improve metabolic health, and possibly slow disease progression in SPG11. Needs careful customization	Dietary intervention; clinical and mechanistic studies
[26]	Calorie-restricted diet characterized by lower energy intake, promoting a decrease in blood glucose, improved mitochondrial function, and reduction of inflammatory activities and neuronal apoptosis.	Could be beneficial in SPG11 by improving mitochondrial efficiency, reducing neuroinflammation, and slowing neurodegeneration.	Dietary intervention; calorie restriction protocol and metabolic impact assessment

## Data Availability

The original contributions presented in this study are included in the article. Further inquiries can be directed to the corresponding author.

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
