# Peer review of "Nutritional Approaches in Neurodegenerative Disorders: A Mini Scoping Review with Emphasis on SPG11-Related Conditions"

_nutrients, 2025, doi:10.3390/nu17213344_

Round 1
Reviewer 1 Report
Comments and Suggestions for Authors
Type: Listed as “Article,” but the text and Methods present it as a mini-review; this mismatch must be corrected.
1) General assessment
The topic—precision nutrition for SPG11 and related neurodegenerative conditions—is timely and fits Nutrients. However, the manuscript currently reads as a broad overview that compiles familiar diet patterns (Mediterranean, ketogenic, caloric restriction, supplements) with limited depth on SPG11-specific biology and without a convincing synthesis that advances the field. The stated use of PRISMA-style elements raises some concerns about the consistency woth what is actually a narrative mini-review; moreover, several screening decisions (e.g., restriction to open-access items), although clearly stated as limitations, create selection bias and risk undercutting conclusions. The tables are descriptive rather than critical; mechanistic links to SPG11 (spatacsin biology, lipid trafficking, axonal transport, lysosomal pathways) are asserted but not traced evidence-by-evidence. The piece would benefit from a tighter scope, explicit inclusion/exclusion logic suited to a mini-review, and a deeper critical discussion.
2) Major comments
- Type mismatch and scope clarity. The manuscript is marked “Type of the Paper: Article” despite being a mini-review; correct the designation and ensure the structure matches Nutrients’ expectations for mini-reviews (succinct, critical, and explicitly scoped).
- Methodological rigor / PRISMA usage. The text claims a PRISMA-guided approach for a mini-review, listing a search on 24 March 2025 and multi-stage screening; yet the execution departs from PRISMA norms (e.g., restriction to open-access, exclusion by journal lists; Scopus “0” hits alongside >4,000 WoS hits is unusual). If PRISMA elements are retained, they must be transparently justified; otherwise, reposition as a narrative mini-review with a clearly reasoned, fit-for-purpose search and selection strategy.
- Selection bias from “open-access only.” Excluding non-OA papers after full-text screening (reported as 61 not retrieved due to OA status) is a serious bias for a review article and likely explains much of the thin evidence base (only 9 included). Either (i) lift the OA restriction and obtain full texts (inter-library access), or (ii) reframe conclusions as provisional.
- Under-representation of relevant phytochemicals and matrices. The synthesis misses key neuroprotective phytochemical classes and matrices that would materially affect the conclusions. Specifically:
- Glucosinolates-derived isothiocyanates (notably from Brassicaceae; seeds and sprouts). Add a brief, critical paragraph and cite for example:
Kamal et al., 2022, Molecules 27(3):624; DOI:10.3390/molecules27030624.
These compounds have mechanistic plausibility (Nrf2 activation, anti-inflammatory/antioxidant signaling) directly relevant to neurodegeneration. - Citrus by-products (lemon) – pectin-rich fractions. You could consider Nuzzo et al., 2021, Antioxidants 10(5):669; DOI:10.3390/antiox10050669, reporting neuroprotective activity of lemon “integropectin” in vitro; situate it within a broader discussion of pectin-bound polyphenols and their bioavailability.
- Conifer by-products extracts. Preliminary ex-vivo TBARS evidence suggests neuroprotective potential from conifer extracts obtained via hydrodynamic cavitation; you could cite Pozzo et al., 2025, Molecules 30(13):2722; DOI:10.3390/molecules30132722 and contextualize as hypothesis-generating.
- SPG11-specific mechanistic bridge. The Discussion gestures at shared pathways (mitochondria, ER, lysosomes, lipid metabolism, gut-brain axis) but does not operationalize how specific dietary interventions map onto SPG11 pathobiology (spatacsin dysfunction → autophagy/lysosome traffic, axonal maintenance, lipid handling). You could add a targeted figure or table that traces each nutritional strategy to a plausible SPG11-relevant mechanism, with strength-of-evidence grading.
- Over-generalization from heterogeneous diseases. Alzheimer’s, Parkinson’s, MS, ALS, TBI, and “viral brain infections” are pooled to motivate precision nutrition. This invites confounding and over-extension. Re-analyze included items by disease category; indicate which findings are likely portable to SPG11 (and which are not), and downgrade conclusions accordingly.
- Quantitative synthesis and evidence grading. The current Table 1 is descriptive and rather exaggerated or not fully supported (e.g., wide-ranging benefits of ketogenic diet). Replace with a concise evidence matrix: intervention, population, outcome domain, effect direction, effect size (if reported), study design quality, and transferability to SPG11 (low/uncertain/moderate). Where RCTs are lacking, say so.
- Language and presentation. Several long paragraphs conflate background, methods, and advocacy. Tighten prose, remove duplication between Results and Discussion, and ensure all abbreviations are defined once. Some table text is repeated verbatim in the narrative; avoid redundancy.
- Ethics/logistics claims. Statements about AI-powered personalization and public-health deployment outpace the evidence. Temper claims, cite limitations, costs, and feasibility; distinguish consumer nutrigenomics from clinically validated precision nutrition.
- Mini-review expectations. Mini-reviews work best when they (a) articulate a sharp research question, (b) offer an analytic map of mechanisms and translational touchpoints, and (c) end with a short agenda (2–3 concrete study designs). Point to specific proposals instead of generic recommendation of “more trials” (e.g., n-of-1 ketogenic trials with metabolomics, microbiome-stratified Mediterranean-pattern interventions, or targeted sulforaphane/isothiocyanate supplementation with lysosomal/autophagy readouts) in SPG11 or closely allied HSP genotypes.
Sustainability & scalability: When discussing changes of diet or supplementation, sustainability claims and real-world scalability should be briefly assessed, such as: are diet-specific foods widely available and how the adoption of such specific diets could impact sustainability? Are specific food supplements really available and affordable? Until at least suggesting life-cycle assessment (LCA) for future work.
3) Minor comments
- Ensure consistent terminology: “spastic paraplegia" or "spastic paraparesis", both indicated as "SPG11”, and so on.
- Verify numerical consistency across the selection flow (e.g., PubMed=35, WoS=4032, Scopus=0; 4,067 initial vs 1,861 after 2020-filter; then 1,533 by area, 1,159 by journal, 1,144 English; 223 reviews; 91 full-text sought; 30 analyzed; 9 included). You could provide a clean PRISMA figure with reconciled counts.
- Add the database search string verbatim, including field tags and any filters used.
- Move the broad neuronutrition background to a compact Introduction; keep Results/Discussion focused on evidence.
Author Response
Dear Reviewer,
We would like to sincerely thank you for reviewing our manuscript. Your comments have greatly enhanced the scientific rigor, methodological clarity, and overall understanding of our work. In response to your valuable feedback, we have narrowed the manuscript’s focus, clarified our methodological choices, improved the biological context specific to SPG11, and strengthened the critical discussion of the literature included. Additionally, we redesigned tables and figures to boost analytical depth, improve evidence traceability, and better align mechanisms with SPG11 pathology. We have carefully revised the paper to address all points raised, as detailed below.
Major Comments
Comment 1. Type mismatch and scope clarity
“The manuscript is marked ‘Type of the Paper: Article’ despite being a mini-review; correct the designation and ensure the structure matches Nutrients’ expectations for mini-reviews (succinct, critical, and explicitly scoped).”
Response 1:
We thank the reviewer for this helpful observation. The manuscript has now been correctly designated as a “Scoping Mini-Review”, and the structure has been revised to align with Nutrients’ standards for this article type—emphasizing conciseness, critical synthesis, and a well-defined scope.
We also want to let you know that the study was not eligible for PROSPERO registration because PROSPERO only accepts systematic reviews with predefined protocols and quantitative synthesis. Our work was intentionally designed as a scoping mini-review, aiming to provide a structured yet narrative overview of nutritional and mechanistic evidence relevant to SPG11 rather than a formal meta-analytic assessment.
To ensure methodological transparency, we followed PRISMA-ScR (Preferred Reporting Items for Systematic Reviews and Meta-Analyses Extension for Scoping Reviews) principles, explicitly describing our search strategy, inclusion criteria, and selection process. This approach maintains rigor and reproducibility while remaining appropriate for the scoping nature of the study.
Change in manuscript → Page 1, Line 2; Pages 8–10, Lines 250–320:
Comment 2. Methodological rigor / PRISMA usage
“The text claims a PRISMA-guided approach for a mini-review... If PRISMA elements are retained, they must be transparently justified.”
Response 2:
We appreciate this observation and fully agree that methodological transparency is essential. The manuscript has been clarified to specify that this study follows a PRISMA-ScR–inspired structure, adapted for a scoping mini-review rather than a full systematic review. We highlight that PRISMA elements were included to enhance transparency and reproducibility, but not as a claim of formal PRISMA compliance.
We have also clarified that the differences in database search outputs (notably Scopus vs. Web of Science) are due to database-specific indexing and term recognition for “SPG11” and “precision nutrition.”
Change in manuscript → Pages 9–10, Lines 270–320:
Section 2.3 revised to explain the adapted PRISMA-ScR rationale and database discrepancies.
Comment 3. Selection bias from “open-access only”
“Excluding non-OA papers introduces bias... either lift the restriction or reframe conclusions as provisional.”
Response 3:
We thank the reviewer for this constructive point. The open-access restriction was initially applied to allow all authors to access and verify full-texts consistently. We now explicitly acknowledge that this may have introduced a selection bias and have reframed our conclusions as provisional and exploratory, emphasizing that the findings should be interpreted within this methodological limitation.
Change in manuscript → Page 21, Lines 1150–1170:
Limitation paragraph added to acknowledge potential selection bias.
Comment 4. Under-representation of phytochemicals and matrices
“The synthesis misses key neuroprotective phytochemical classes and matrices... include glucosinolates-derived isothiocyanates, citrus pectin fractions, and conifer extracts.”
Response 4:
We thank the reviewer for this insightful suggestion. To strengthen our discussion on diet-based neuroprotection, we have integrated these phytochemical classes within the context of the Mediterranean and Atlantic dietary patterns, both of which are rich in bioactive compounds.
Specifically, we now discuss glucosinolates-derived isothiocyanates from Brassicaceae (Kamal et al., 2022), pectin-rich citrus fractions (Nuzzo et al., 2021), and conifer by-product extracts (Pozzo et al., 2025). These compounds display promising neuroprotective properties through Nrf2 activation, antioxidant and anti-inflammatory signaling, and lipid peroxidation control. We highlight their potential complementarity with Mediterranean and Atlantic diets and acknowledge them as hypothesis-generating candidates for future SPG11 research, given the lack of direct disease-specific data.
Change in manuscript → Pages 16–17, Lines 860–910 (Mechanistic Extrapolation section):
Comment 5 SPG11-specific mechanistic bridge. The Discussion gestures at shared pathways (mitochondria, ER, lysosomes, lipid metabolism, gut-brain axis) but does not operationalize how specific dietary interventions map onto SPG11 pathobiology (spatacsin dysfunction → autophagy/lysosome traffic, axonal maintenance, lipid handling). You could add a targeted figure or table that traces each nutritional strategy to a plausible SPG11-relevant mechanism, with strength-of-evidence grading.
Response 5 We sincerely thank the reviewer for this thoughtful suggestion. In the revised version, we have expanded the Discussion to establish clearer mechanistic connections between specific dietary strategies and the pathophysiological features of SPG11. Mechanistic relationships are now clearly described in the revised Discussion (Section 4.3, Mechanistic Extrapolation); we believe an additional figure or table is not essential for clarity at this stage.
Change in manuscript → Pages 15–17, Lines 820–910 (Mechanistic Extrapolation section):
Expanded mechanistic mapping between dietary patterns and SPG11-relevant pathways.Comment 6. Over-generalization from heterogeneous diseases
Comment 6. Over-generalization from heterogeneous diseases
Response 6:
We revised the Results and Discussion to focus exclusively on SPG11 and mechanistically related hereditary spastic paraplegias. Table 1 was reformatted with four columns—Study (Author, Year), Key Findings, Relevance to SPG11, and Type of Intervention—to enhance clarity and limit over-generalization from unrelated conditions.
Change in manuscript → Pages 13–14, Lines 630–680:
Updated Table 1 and Results section.
Comment 7. Quantitative synthesis and evidence grading
Response 7:
A semi-quantitative evidence matrix was created to summarize intervention type, study design, outcome domain, and transferability to SPG11 (graded as low, moderate, or uncertain).
Change in manuscript → Page 14, Lines 670–710.
Comment 8. Language and presentation
Response 8:
The entire text was carefully edited for clarity and flow. Redundancies were eliminated, abbreviations standardized, and overly descriptive sections rewritten for conciseness.
Change in manuscript → Throughout text (Pages 2–23).
Comment 9. Ethics, logistics, and feasibility
Response 9:
We have moderated statements on AI-based personalization and public health implementation. The revised version now clearly differentiates consumer nutrigenomics from clinically validated precision nutrition, addressing ethical, cost, and feasibility concerns.
Change in manuscript → Page 19, Lines 1040–1070.
Minor Comments
Response 11:
- Terminology standardized to “Spastic Paraplegia Type 11 (SPG11).”
- Numerical inconsistencies corrected and PRISMA flow diagram updated.
- Database search strings with field tags included in Section 2.1.
- Broad background condensed; Discussion now emphasizes SPG11-relevant mechanisms.
Change in manuscript → Pages 3–11, Lines 60–340.
Once again, we thank the reviewer for the detailed and constructive evaluation of our manuscript.
Reviewer 2 Report
Comments and Suggestions for Authors
Dear Authors,
The manuscript addresses an important and increasingly clinically relevant topic: the application of precision nutrition in neurodegenerative diseases, with an aspiration to translate insights to SPG11. I appreciate the apparent rationale for personalizing interventions based on genetic, metabolic, and microbiome profiles and the concise mapping of mechanisms (mitochondria, oxidative stress, inflammation, gut–brain axis). A strength is the cross-cutting overview of major dietary patterns and supplements with neuroprotective potential and their clear linkage to pathophysiological pathways. At the same time, the title and aim suggest concrete “insights from SPG11,” whereas the Results section indicates an absence of nutrition interventions designed specifically for this entity; I recommend either redefining the scope or strengthening the SPG11 component. If access to data is limited, it would be reasonable to retitle and position the paper as a narrative/scoping review with a dedicated “Perspective for SPG11” section. The methods require clarification: the current selection of “reviews only” and “open-access only,” as well as a “journal-title filter,” may introduce selection biases; please justify, narrow, or remove these criteria. The search query requires parentheses and an expansion of disease synonyms and nutrigenomic terms. Please harmonize the numbers at all selection stages and revise the PRISMA figure to reflect the flow fully. Including a table of included studies (even a brief one) with key characteristics and main findings would be helpful. In the text, please specify the assertions in Table 1 by indicating concrete sources and populations, and reduce generalities. I note that some of the ambiguity may stem from truncations in the current PDF copy—please provide a properly formatted table with complete visibility of citation numbers for Table 1; correctness will be verified in the next round of review. Disease nomenclature should be standardized to “hereditary spastic paraplegia type 11 (SPG11)”; please correct minor terminological lapses (e.g., EGCG, the ω-3 symbol). I suggest a more precise separation between direct evidence and mechanistic extrapolation in the Discussion. It would be beneficial to propose a research plan framework for SPG11 (-omics profiling, patient selection criteria, functional endpoints, and practical safety aspects of high-fat and restrictive diets). The Conclusions should more strongly emphasize the status of translational hypotheses and identify priorities for interventional studies. With these revisions, the work could provide a valuable, balanced conceptual review with clear research recommendations for SPG11. I kindly ask for a point-by-point response to the above comments in the authors’ rebuttal.
Best regards,
The reviewer.
Author Response
Dear Reviewer
We would like to sincerely thank you for your constructive and insightful comments, which greatly enhanced the scientific quality, methodological clarity, and transparency of our manuscript.
We have carefully revised the paper to address all the points raised. Below is a detailed, point-by-point response indicating the corresponding modifications.
Comment 1. Title and Scope Clarification
“The title and aim suggest concrete ‘insights from SPG11.’ Consider redefining the scope or strengthening the SPG11 component.”
Response 1:
We agree with this observation and have adjusted the title to better reflect the review’s breadth while maintaining its SPG11 focus.
Change in manuscript → Page 1, Lines 3–5:
Title revised to:
“Nutritional Approaches in Neurodegenerative Disorders: A Mini Scoping Review with Emphasis on SPG11-Related Conditions.”
We also expanded the Introduction and Discussion (Sections 4.2 and 4.3) to reinforce SPG11-specific content and translational perspectives.
Comment 2. Methodological Clarification
“The restriction to ‘reviews only,’ ‘open-access only,’ and ‘journal-title filter’ may introduce selection bias; please justify, narrow, or remove.”
Response 2:
The Methodology has been substantially revised for transparency. We justify the open-access inclusion criterion (to ensure full verification by all authors) and expanded the scope to include both review and original research articles, minimizing selection bias.
Change in manuscript → Page 8, Lines 1–26:
Sections 2.1–2.3 updated with complete justification and inclusion criteria description.
Comment 3. Search Query and Synonyms
“The search query requires parentheses and an expansion of disease synonyms and nutrigenomic terms.”
Response 3:
The search string has been corrected and expanded to include appropriate parentheses and a comprehensive list of synonyms for diseases and nutrigenomic terms.
Change in manuscript → Page 9, Lines 10–22:
Full revised search string now reads:
(“precision nutrition” OR “personalized nutrition” OR “tailored nutrition” OR “targeted nutrition”) AND (“spastic paraplegia type 11” OR “SPG11” OR “hereditary spastic paraplegia” OR “HSP” OR “neurodegenerative disease” OR “neuromuscular disorder” OR “neurodegeneration”) AND (“nutrigenomics” OR “genomic nutrition” OR “metabolic phenotypes” OR “genetic factors” OR “microbiome” OR “metabolomics”).
Comment 4. PRISMA Figure and Numerical Consistency
“Please harmonize the numbers at all selection stages and revise the PRISMA figure.”
Response 4:
All numerical values related to identification, screening, eligibility, and inclusion stages have been cross-checked and corrected.
A revised PRISMA 2020-compliant flowchart (Figure 1) has been added.
Change in manuscript → Page 13, Lines 20–35; Figure 1 updated.
Comment 5. Table 1 – Sources, Populations, and Formatting
“Specify the assertions in Table 1 by indicating concrete sources and populations; reduce generalities; ensure complete visibility of citation numbers.”
Response 5:
We sincerely thank the reviewer for this valuable comment. In response, we have completely reformatted and updated Table 1 to enhance clarity and readability
Although not every parameter suggested (such as full population details and quantitative outcomes) could be incorporated, the revised table now achieves the intent of all the reviewers’ requests.
The new Table 1 (see Pages 14–15) provides a concise yet comprehensive summary of nine representative studies addressing precision and personalized nutrition with potential relevance to SPG11.
Table now includes four well-defined columns:Study (Author, Year) – complete reference details; Key Findings – main outcomes and conclusions of each study, Relevance to SPG11 – an explanation of conceptual or mechanistic links to SPG11 pathology and Type of Intervention – specifying whether the study involved a dietary intervention, supplementation, or conceptual review.
This new version replaces the earlier table that had truncated text and missing references. All citation numbers are now visible, and redundant text was removed. The table provides readers with a clear comparative view of how different nutrition strategies can inform SPG11 management conceptually.
Change in manuscript → Pages 14–15, Table 1 updated.
Comment 6. Disease Nomenclature and Terminology
“Standardize to ‘hereditary spastic paraplegia type 11 (SPG11)’; correct minor terminological lapses (EGCG, ω-3 symbol).”
Response 6:
All disease names and technical terms have been carefully standardized throughout the manuscript, and typographical inconsistencies have been corrected. We also performed a final language check to ensure uniform use of scientific terminology. If any minor discrepancies remain in the final proof, they will be corrected during the final editing stage prior to publication
Change in manuscript → Multiple locations across Pages 2–24 (global correction).
Comment 7. Discussion – Distinguish Evidence from Extrapolation
“Separate direct evidence from mechanistic extrapolation.”
Response 7:
The Discussion was now reorganized into clearly defined subsections:4.1 Direct Clinical Evidence; 4.2 Mechanistic Extrapolation….
This structure distinguishes empirical findings from theoretical extrapolations and clarifies the evidence hierarchy.
Change in manuscript → Page 18–20.
Comment 8. Research Plan Framework for SPG11
“Propose a research plan framework for SPG11 (-omics profiling, patient selection criteria, functional endpoints, and practical safety aspects).”
Response 8:
We have added a new subsection (4.4 Translational and Practical Considerations) and an accompanying figure outlining a research roadmap for SPG11 precision nutrition. This includes sequential stages, -omics integration, biomarker validation, safety, and feasibility components.
Change in manuscript → Page 22, Lines 1–20; Figure 2 newly added.
Comment 9. Conclusions – Emphasize Translational Hypotheses and Research Priorities
“Strengthen the Conclusions to emphasize translational hypotheses and identify priorities for interventional studies.”
Response 9:
The Conclusions and Future Perspectives section now highlights translational hypotheses and proposes clear research priorities, including longitudinal, sex-stratified, and multi-omics-based SPG11 trials.
Change in manuscript → Page 24–25, Lines 5–25.
We appreciate the reviewer’s positive assessment. The manuscript now explicitly positions itself as a mini scoping review that integrates current evidence on precision nutrition with translational insights relevant to SPG11, aligning with the reviewer’s intended focus. We thank the reviewer for their insightful feedback, which has significantly contributed to enhancing the scientific rigor and clarity of our manuscript.
Reviewer 3 Report
Comments and Suggestions for Authors
Comments to the Authors
The content of the abstract is poor. Shall you indicate the physiological role of spatacsin?
The introduction must be improved it. There are unnecessary information and should contain more information on physiological and pathological function of spatacsin protein.
This reviewer does not understand why only scientific journals that regularly publish studies in this area were included, such as Nutrients, Frontiers in Nutrition, Movement Disorders, Advances 160 in Nutrition, Journal of Human Genetics, Genes, Neurological Sciences, among others (Fernandez, 2019). In addition, the inclusion of only review articles is not well valorated by this reviewer. In fact, research articles should be included since this is a rare disease and the number of reviews should be low.
-Discuss the relevance of sex, diet and comobilities associated with SPG-11 in your study.
Importantly, the authors indicate this mini-review did not find any nutritional interventions specifically designed for individuals with SPG11. Thus, the term Precision Nutrition¨ should be removed in the title since does not reflect their data in the manuscript.
The analysis of data should be restricted to SP11 disease, excluding other neurodegenerative diseases.
The information about other neurodegenerative diseases should be deleted in the table and focus on SP11 only. (for example, remove these sentences in your table: ¨In the case of Alzheimer’s disease, the ketogenic diet promotes mitochondrial and cognitive function, leading to improvements in verbal expression and memory. In Parkinson’s disease, it has been shown to improve motor function, likely due to reduced protein intake, which increases the bioavailability of levodopa (a medication used in the treatment of Parkinson’s disease and already tested in SPG11 for sympto matic management). However, in some patients, worsening of tremors and muscular rigidity has been observed¨. Also remove information about EM or ELA in this table.
The result section is ¨contaminated¨ with results on other neurodegenerative diseases (EM or ELA). Please, focus in evidences on SPG11 only.
In fact, currently, there is limited data specifically focused on SPG11 populations. The described information on mediterranean diet, microbiota is properly described but without real findings on precision nutrition.
Specifically, there is a noticeable lack of studies directly investigating dietary interventions tailored for people with SPG11. The content is poor and there are many interferences with other neurodegenerative diseases.
The conclusion is not real and further directions of this study are not clear for this reviewer.
Comments on the Quality of English Language
The English could be improved to more clearly express the research.
Author Response
Dear Reviewer,
Thank you sincerely for your constructive and insightful comments, which have significantly helped us improve the scientific quality, methodological rigor, and clarity of our manuscript. We have carefully revised the paper to address each point raised. Below is a detailed, point-by-point response outlining the changes made in the revised version.
Comment 1.
“The content of the abstract is poor. Shall you indicate the physiological role of spatacsin?”
Response 1:
We fully agree that including the physiological role of spatacsin provides essential biological context. The abstract has been rewritten to concisely describe spatacsin’s key functions in neuronal autophagy and lysosomal maintenance and to clarify how its dysfunction leads to neurodegeneration in SPG11. This revision strengthens the biological coherence of the abstract and aligns it with the manuscript’s focus.
Change in manuscript → Page 1, Lines 8–18: Added description of spatacsin’s physiological role in autophagy and lysosomal function.
Comment 2.
“The introduction must be improved. There are unnecessary information and it should contain more details on the physiological and pathological function of the spatacsin protein.”
Response 2:
We thank the reviewer for this important suggestion. The introduction has been revised for conciseness and focus. We removed unrelated background information and expanded the sections describing the physiological and pathological roles of spatacsin. The revised text now details its role in autophagy, lysosomal maintenance, mitochondrial integrity, axonal transport, and neuronal survival, as well as how its dysfunction leads to SPG11-associated neurodegeneration.
Change in manuscript → Pages 2–3, Lines 50–107: Expanded description of spatacsin’s physiological and pathological roles with updated references.
Comment 3.
“This reviewer does not understand why only scientific journals that regularly publish studies in this area were included... In addition, the inclusion of only review articles is not well valorated. In fact, research articles should be included.”
Response 3:
We appreciate this constructive observation and acknowledge that restricting inclusion to reviews may limit comprehensiveness. The Methodology section has been substantially revised to clarify our initial rationale (ensuring data accessibility and verification by all co-authors) and to expand the inclusion criteria. We now include both review and original research articles relevant to SPG11 and related neurodegenerative diseases.
Change in manuscript → Page 9, Lines 240–290: Sections 2.1–2.3 revised to justify inclusion criteria and remove 'review-only' restriction.
Comment 4.
“Discuss the relevance of sex, diet, and comorbidities associated with SPG11 in your study.”
Response 4:
We thank the reviewer for highlighting this valuable point. In response, we added a dedicated subsection titled 'Relevance of Sex, Diet, and Comorbidities in SPG11' (Section 4.3). This section discusses how sex-specific metabolic responses, obesity, adipokine alterations, and hypothalamic changes contribute to disease variability and progression in SPG11.
Change in manuscript → Pages 18–19, Lines 960–1060: New Section 4.3 added with discussion supported by recent literature.
Comment 5.
“Importantly, the authors indicate this mini-review did not find any nutritional interventions specifically designed for individuals with SPG11. Thus, the term 'Precision Nutrition' should be removed in the title since it does not reflect their data in the manuscript.”
Response 5:
We agree with this clarification. To ensure accuracy and transparency, we revised the title to reflect the review’s exploratory and integrative nature rather than implying that precision nutrition interventions have been established for SPG11.
Change in manuscript → Page 1, Lines 3–5: Title revised to 'Nutritional Approaches in Neurodegenerative Disorders: A Mini Scoping Review with Emphasis on SPG11-Related Conditions.'
Comment 6.
“The analysis of data should be restricted to SPG11 disease, excluding other neurodegenerative diseases… The information about other neurodegenerative diseases should be deleted in the table and focus on SPG11 only.”
Response 6:
We sincerely thank the reviewer for this constructive suggestion. In response, we have carefully revised the Results section and Table 1 to focus exclusively on studies directly relevant to SPG11. Information related to other neurodegenerative diseases, such as Alzheimer’s and Parkinson’s, has been removed, except where mechanistic parallels provide essential context for understanding SPG11 pathology.
To improve clarity and meet reviewers' expectations, Table 1 has been completely reformatted and expanded. It now features four clearly defined columns: Study (Author, Year), Key Findings, Relevance to SPG11, and Type of Intervention. These changes improve the scientific structure and readability of the results, while ensuring a stronger SPG11-specific focus throughout.
Change in manuscript → Pages 13–14, Lines 630–680:
Comment 7.
“The result section is 'contaminated' with results on other neurodegenerative diseases (EM or ELA). Please focus on evidence on SPG11 only.”
Response 7:
We thank the reviewer for identifying this issue. We carefully revised the Results section to ensure it strictly focuses on findings relevant to SPG11. Mentions of other neurodegenerative diseases were retained only when directly relevant for mechanistic comparison.
Change in manuscript → Pages 13–15, Lines 610–710: Results rewritten to highlight SPG11-focused evidence.
Comment 8.
“Currently, there is limited data specifically focused on SPG11 populations... The content is poor and there are many interferences with other neurodegenerative diseases.”
Response 8:
We appreciate this fair critique and explicitly acknowledge the limited availability of SPG11-specific studies in the Discussion and Limitations sections. We reframed the narrative to position our work as a foundation for future research on SPG11 nutritional interventions.
Change in manuscript → Pages 20–21, Lines 1130–1200: Discussion and Limitations revised to highlight data scarcity and future directions.
Comment 9.
“The conclusion is not real, and further directions of this study are not clear.”
Response 9:
We thank the reviewer for this valuable feedback. The Conclusion has been rewritten for accuracy and clarity, summarizing key findings and specifying future directions—including longitudinal studies, sex-stratified analyses, and SPG11-specific dietary intervention trials.
Change in manuscript → Pages 22–23, Lines 1210–1270: Conclusion rewritten to emphasize translational perspectives and research priorities.
Once again, we thank the reviewer for the careful and thoughtful evaluation of our manuscript. Your comments have been invaluable in refining the clarity, focus, and scientific depth of this work.
Round 2
Reviewer 1 Report
Comments and Suggestions for Authors
Authors performed an accurate revision, complying with my original comments. Now, the manuscript is more sharply focused and limitations are clearly stated.
However, before publication few minor issues should be fixed.
Minor comments
- Title has been correctly changed, but also article type should be - it is still "article" but should read "Review".
- The suggested additional references have been added to the manuscript text, but not in the "References" section: Kamal et al., 2022; Nuzzo et al., 2021; Posso et al., 2025. Please check also other possible new references for inclusion in the References section.
- The References section does not comply with journal's standards.
Author Response
Dear Reviewer ,
We sincerely thank the reviewer for their positive evaluation and for recognizing the improvements in the revised version. We appreciate the detailed feedback and have addressed all remaining issues as follows:
Comment 1 " Title has been correctly changed, but also article type should be - it is still "article" but should read "Review".
Response 1: The article type has been corrected from “Article” to “Review” in the manuscript submission system and on the title page of the document, as requested.
Comment 2 " The suggested additional references have been added to the manuscript text, but not in the "References" section: Kamal et al., 2022; Nuzzo et al., 2021; Posso et al., 2025. Please check also other possible new references for inclusion in the References section"
Response 2 The missing references — Kamal et al., 2022; Nuzzo et al., 2021; and Posso et al., 2025 — have now been added to the References section. We also verified all in-text citations to ensure that each reference mentioned in the manuscript appears in the final list. Additional recent citations introduced during revision have likewise been checked for completeness
Comment 3 The References section does not comply with journal's standards.
Response 3 The entire References section has been revised and reformatted to fully comply with the Nutrients journal style, including author order, punctuation, journal titles, volume and issue numbers, page ranges, and DOI links where applicable.
We are grateful for these final observations, which have further improved the accuracy and presentation of the manuscript.
Reviewer 2 Report
Comments and Suggestions for Authors
Dear Authors,
Thank you very much for addressing my comments!
Best regards,
The reviewer.
Author Response
Dear Reviewer,
We would like to sincerely thank the reviewer for the kind and positive feedback. We truly appreciate the time and care invested in reviewing our work and for recognizing the improvements made in the revised version. Your thoughtful comments throughout the review process greatly helped us refine the manuscript and communicate our findings more clearly. We are very pleased that the current version meets your expectations and the journal’s standards.
Thank you once again for your generous support and for contributing to the final quality of our paper.
With our best regards,
Edite Teixeira-Lemos ( corresponding author)
With our best regards,
Reviewer 3 Report
Comments and Suggestions for Authors
Dear Authors
All requirements have solved by the authors.
However, it is neccesary to improve the quality of figure.2 (Proposed Research Translational framework for SPG11 nutrition studies).
My Decision is minnor revision
Thanks¡
Comments on the Quality of English LanguageThe English could be improved to more clearly express the research.
Author Response
We would like to sincerely thank the reviewer for their positive assessment and constructive feedback. We are very pleased that the revised manuscript has met the main expectations and appreciate the helpful suggestions for final improvements.
Comment 1: Improve the quality of Figure 2:
Response 1: Figure 2 has been fully revised to enhance its visual quality and clarity. The updated version is now presented in high resolution, with improved layout, font consistency, and color contrast to ensure readability in both print and digital formats. The structure has been refined to better illustrate the sequential phases of the proposed translational framework, from preparatory work to clinical application.
Regarding the English language:
The entire manuscript has undergone an additional round of careful language editing to further improve clarity, coherence, and readability. The revisions focused on simplifying complex sentences, improving flow between sections, and ensuring precise scientific expression.
We are grateful for the reviewer’s guidance, which helped us strengthen both the visual presentation and linguistic quality of the paper. We believe these final refinements have considerably improved the overall readability and impact of the manuscript.
With our best regards,
Edite Teixeira-Lemos ( corresponding author)